# Study of Innovative Connector for Steel–Concrete Composite Structures

Anna Derlatka [1,*], Piotr Lacki [1], Paweł Kania [1] and Shan Gao [2]

1 Faculty of Civil Engineering, Czestochowa University of Technology, Dabrowskiego 69, 42-201 Czestochowa, Poland; piotr.lacki@pcz.pl (P.L.); pawel.kania@pcz.pl (P.K.)
2 Key Lab of Structures Dynamic Behavior and Control of the Ministry of Education, Harbin Institute of Technology, 73 Huanghe St., Nangang, Harbin 150001, China; 13833185232@139.com
* Correspondence: anna.derlatka@pcz.pl

**Featured Application: The proposed solution of fasteners made of corrugated steel sheet and shot nails can be used as the connector for steel–concrete composite floor beams. The sheet provides stay-in-place formwork for a monolithic reinforced concrete slab. Furthermore, the sheet in the shape of the dovetail caused the connection between the steel I-beam and the concrete slab. The connectors are easy to fabricate. Any special mounting equipment that requires high-current electrical power is not necessary. Because shot nails have very wide applications, nailing devices are available in almost every construction factory and on construction sites. The shape of a such cross-section of the metal sheet improves its stability during installation compared to a flat metal sheet. An additional advantage is the ability to easily suspend the finishing elements by using dedicated hangers fixed in the fold of sheet.**

**Abstract:** The paper presents an analysis of the innovative connector for manufacturing a steel–concrete composite beam. The connector consists of a corrugated metal sheet in the shape of a dovetail and shot nails. The nails are shot through the sheet fold into the flange of the steel I-section. Experimental studies of push-out tests were carried out. The conducted tests proved that the proposed solution can be applied as the fastener for steel–concrete composite beams for the construction of ceilings in utility public buildings with small beam's span. Considering the criteria presented in the Eurocode 4 standard and the results of the experiments, it was proven that all analyzed types of fasteners are ductile. The connector made of sheet with a thickness of 1.00 mm and 2 nails is characterized by a breaking load of 30.83 kN. The load-bearing capacity of the fastener can be adjusted by changing the corrugated sheet thickness and changing the number of nails shot in the single fold of the sheet.

**Keywords:** steel–concrete composite structures; connector; push-out test; corrugated sheet; ceiling construction



## 1. Introduction

The most popular structural solution of the steel–concrete composite beams are beams with slabs concreted on profiled steel sheets. The sheet metal, apart from cooperating with concrete in the use phase of the slab, is also a working platform and formwork lost in the execution phase [1–3]. From a structural point of view, the slab serves as reinforcement for positive bending moments. In composite beams, it is important to ensure the proper connection of the steel and concrete parts. In bent elements, the natural adhesion occurring between the steel section and the concrete slab does not ensure the transfer of forces that appear at the contact area of the two materials. The purpose of the fasteners is to transfer the longitudinal delamination forces arising at the contact surface and to prevent the slab from detaching from the steel beam [4,5].

To this day, in building construction, headed studs are most often used to join the slab with the steel beam [6–10]. They absorb both shear and tensile forces, and their load

capacity is the same in all directions. In addition, they do not constitute a significant obstacle to the reinforcement of the slab. As shown in [11,12], note that, in solid slabs, the pin acts as a cantilever. In plates on corrugated sheets, the greatest loads act on the upper part of the stud, protruding above the plane of the upper corrugation of the sheet. In this area, the stud presses against the concrete and causes it to crush. However, on the other side of the stud, a crack in the concrete appears.

Despite the many known methods of connection used in the composite structures, research is still being carried out to create new solutions. The authors of the article [3] presented the possibility of connecting the steel beam with the slab using a non-welded hat connector. Connectors made of the hat section 80 × 85 × 3 made of S235 steel were fastened to the beam with four driven nails and a diameter of ϕ = 4.5 mm. The test results showed that a fastener made of a 100 mm long sheet corrugation cannot be used because its slip capacity is only 2.6 mm. A connector made of a 60 mm long sheet corrugation can be used to join steel–concrete beams, as its slip is 7.1 mm.

An innovative type of shear connector was also presented by the authors of [13]. Instead of shear studs, a continuous hat channel was used. The structural system consisted of: steel beams made of two cold-formed profiles, corrugated steel sheet, hat channels, and the concrete slab with transverse reinforcement. Self-drilling fasteners were used to connect all steel parts. The authors of the paper [13] claimed that the hat channel was able to effectively transfer the shear between the steel and the concrete to achieve the connection.

The Hilti Stripcon connector made of the corrugated sheet connecting the beam with the slab on the corrugated sheet is shown in [1]. The connector is made of an 80 mm wide S280GD steel sheet. In each corrugation of the sheet, the connector is fastened with four Hilti driven nails. In addition, to increase the mechanical connection, holes were used in the connector plate. A characteristic feature of this solution is the greater ability to slip compared to headed studs.

A completely different approach to joining the steel profile with the prefabricated reinforced concrete slab is presented in the works [14–16], where glue was used as a connector. Experimental results show that it is possible to make a composite structure glued with glue. The connection provided by the epoxy adhesive is non-slip at the steel–concrete interface. In the case of joining with polyurethane glue, the connection is flexible. The failure of the composite beams was caused by the plasticization of the steel girder or cracking of the concrete slab.

One of the solutions that deserves special attention is the system of connectors made of a properly shaped steel beam web, the so-called "composite dowels". The solution of a composite beam made of the reinforced concrete slab connected to an I-beam with a properly cut web is known from bridge structures, as presented in [17,18]. Therefore, they can also be used on heavily loaded industrial or warehouse ceilings [2]. As presented in [19], the main advantage of the "composite dowel" compared to head studs is a higher load capacity and adequate resistance to deformation even in high-strength concrete, thanks to which, according to EN 1994-1-1 [20], they can be classified as ductile shear connectors.

The test results of an innovative connector in the form of a steel sheet with an embossed steel plate are presented in [21]. The fasteners are welded to the steel beam. The proposed connector is very durable due to the strength of bonding with concrete. The stiffness of this connector turned out to be greater than that of the headed studs. At the same time, the results of the experiment showed that the slip ability of the sheet metal fastener was also greater than that of the stud. In addition, the sheet metal connection turned out to be less prone to cracking as the embossing pattern used is very durable and resistant to dynamic loads.

For mechanical connection, a popular design solution is the use of trapezoidal sheet metal with embossing [22,23]. Most of them, in the upper part of the fold of the sheet, have concave ribbing so that the concrete can connect with the sheet [8]. There are also solutions in which the upper part of the fold uses convex embossing, so that the sheet can be embedded in the concrete.

An innovative alternative to trapezoidal sheets was presented in [24], where the sheet with a sinusoidal cross-section was used. Experimental tests, as well as numerical simulations, confirm the possibility of using such a sheet shape. However, the sinusoidal cross-section prevents the welding of the studs. Therefore, the authors of [24] designed a dedicated connector, which is much heavier than a headed stud.

In the steel–concrete composite structures, the friction joint is created using specially shaped sheet folds. As presented in the works [22,25], such a shape of the cross-section improves the stability of the sheet and also acts as hooks connecting to the concrete, preventing the global buckling of the slab. Among the advantages of slabs on all types of corrugated sheets, there is no need to make formwork, because the sheet performs this function.

This article proposes a novel connector for manufacturing the steel–concrete composite beam. The connector consists of shot nails and corrugated sheet in the shape of the dovetail. Nails are driven through the bottom fold of the sheet into the I-beam flange. The sheet provides stay-in-place formwork for a monolithic reinforced concrete slab, and its shape affects the connection of the reinforced concrete slab with the steel I-beam. The experimental tests of the shear load capacity of the fasteners were carried out. Three types of fasteners, differentiated in sheet thicknesses and the number of nails driven into the sheet fold, were analyzed. The research aims to verify the thesis of whether it is possible to provide the steel–concrete connection without the use of additional connecting components. In the proposed solution, the most popular type of connector, i.e., head studs, were eliminated. The connection was achieved only using a properly formed corrugated sheet. At the same time, these are the first tests of this type of connector. In the future, it is planned that a numerical model of this type of connection as well as parametric analysis and the optimization of the solution will be developed.

## 2. Materials and Methods

To make fasteners for composite structures, it was proposed to use the corrugated sheet and the shot nail with the shapes shown in Figure 1. The proposed dimensions of the sheet are shown in Table 1. However, Table 2 contains the minimum requirements for the nail. After researching the market in terms of the availability of sheets and nails corresponding to the proposed criteria, it was decided to perform the push-out tests of fasteners made of corrugated sheet steel grade S280GD marked in accordance with EN 10346:2015-09 [26], covered on both sides with a zinc coating and nails with a diameter $\phi = 4.5$ mm and the length of the shot-in part $h_n$ in the range of 16.0–17.5 mm, made of carbon steel protected against corrosion. The choice of material from which the sheet metal and nails were made and their dimensions were dictated by their availability on the local market.

**Table 1.** Proposed dimensions of the corrugated sheet (mm).

| $t_f$ | $h_f$ | $m$ | $f_l$ | $f_u$ | $s$ | $R$ | $L_b$ |
|------|------|-----|------|------|-----|-----|------|
| 0.50 | 59 | 140 | 127 | 44.6 | 13 | 5 | 560 |
| 0.80 | 59 | 140 | 127 | 44.6 | 13 | 5 | 560 |
| 1.00 | 59 | 140 | 127 | 44.6 | 13 | 5 | 560 |
| 1.25 | 59 | 140 | 127 | 44.6 | 13 | 5 | 560 |

**Table 2.** Proposal of minimum requirements for the shot nail.

| Criterion | Minimum Requirement |
|-----------|---------------------|
| Diameter $\phi_n$, mm | 4.5 |
| Length of the shot part $h_n$, mm | 16.0 |
| Yield strength, MPa | 950 |
| Tensile strength, MPa | 2000 |

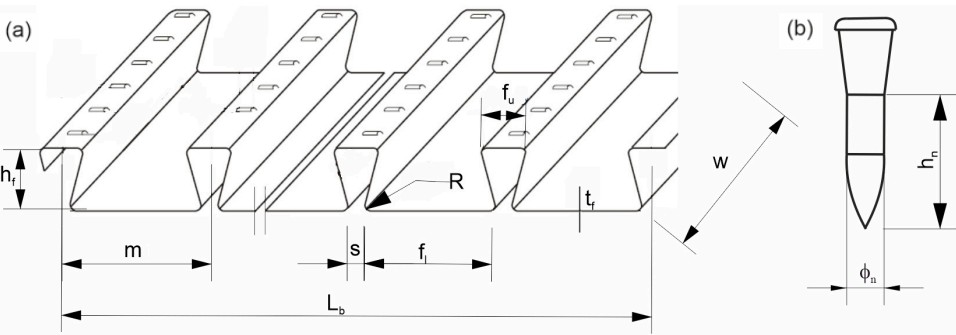

**Figure 1.** Shape: (**a**) corrugated sheet; and (**b**) shot nail.

Experimental tests were carried out on the raw material (steel I-beams, steel of corrugated sheets, and concrete) and innovative fasteners for a steel–concrete composite structures made of corrugated metal sheet 1.0 mm thick and 4 nails, corrugated metal sheet 1.0 mm thick and 2 nails, and corrugated steel 1.25 mm thick and 2 nails.

*2.1. Material Properties*

In order to verify the properties of the steel grades used, the static tensile tests were performed for samples cut from:

- HEA 160 I-beam made of S235JR steel grade marked in accordance with EN 10025-2 [27], used to make samples for push-out test of connectors;
- The corrugated sheets, 1.00 and 1.25 mm thick, used to make samples for testing connectors and to make composite beam.

The geometries of the flat samples used for the static tensile tests of the material of the HEA 160 sections are shown in Figure 2a. The geometry of the flat samples used to determine the material properties of the corrugated sheets is shown in Figure 2b. The geometries were developed on the basis of EN ISO 6892-1 [28]. In order to determine the repeatability of the results, the study was carried out on three samples cut from the web of HEA 160 I-beam and three samples cut from each corrugated sheet with a thickness of 1.00 and 1.25 mm.

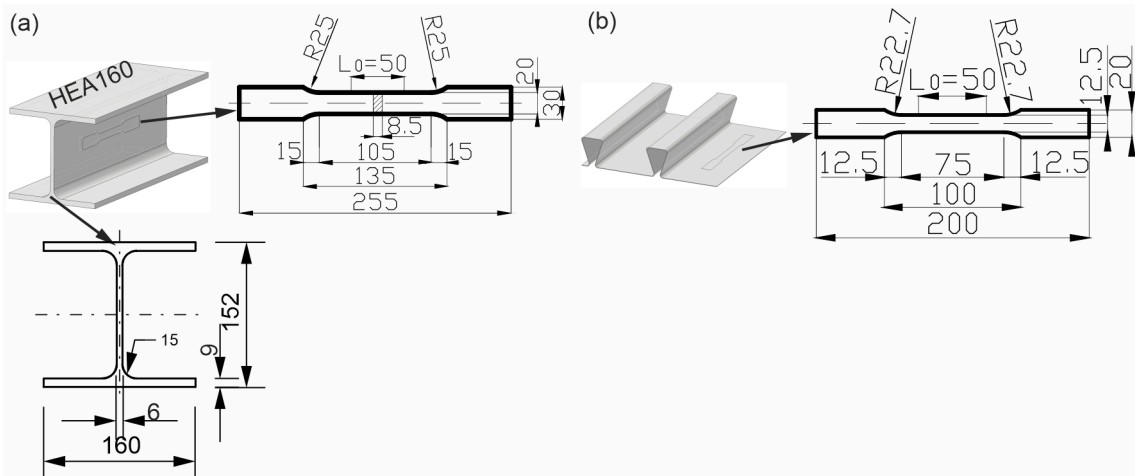

**Figure 2.** Shape of sample for testing the material properties of: (**a**) I-beam; and (**b**) corrugated sheets.

All concrete components were made of the C20/25 concrete class marked in accordance with EN 206+A2 [29]. In order to verify the mechanical properties of the concrete, tests of the compressive strength of the concrete used for the execution of samples for the push-out tests of the connectors for composite structures were performed. For the classification of concrete, the characteristic compressive strength determined after 28 days of curing on the

cubic samples with a side of 150 mm marked $f_{ck.cube}$ in accordance with EN 206+A2 [29] was used.

### 2.2. Push-Out Test

The subject of the research was models for the shear testing of connectors for composite structures. The innovative connector between the steel I-beam and the reinforced concrete slab is the corrugated sheet in the shape of a dovetail attached to the I-beam with shot nails. Nails were driven into the lower part of the fold, in contact with the I-beam. Samples (models) made of sheet metal with a thickness of 1.00 and 1.25 mm and with 4 and 2 nails in the single fold of the sheet were analyzed.

Since the EN 1994-1-1 [20] standard specifies the requirements for the geometry of standard test specimens only for typical headed studs, special tests were carried out in accordance with the guidelines [20]. A diagram of the construction of the model for the push-out test of the connectors is shown in Figure 3. Each model was built of the HEA 160 I-beam made of S235 structural steel, 500 mm long. The sheets, galvanized on both sides, were fastened to the I-section flanges. Grease was applied between the HEA profile and the metal sheet to eliminate adhesion on the joint surface. Corrugated sheets made of S280GD steel with a width of 400 mm and a length of 500 mm were used. The corrugated sheet, in addition to the connector function, also provided permanent formwork for the monolithic reinforced concrete slab.

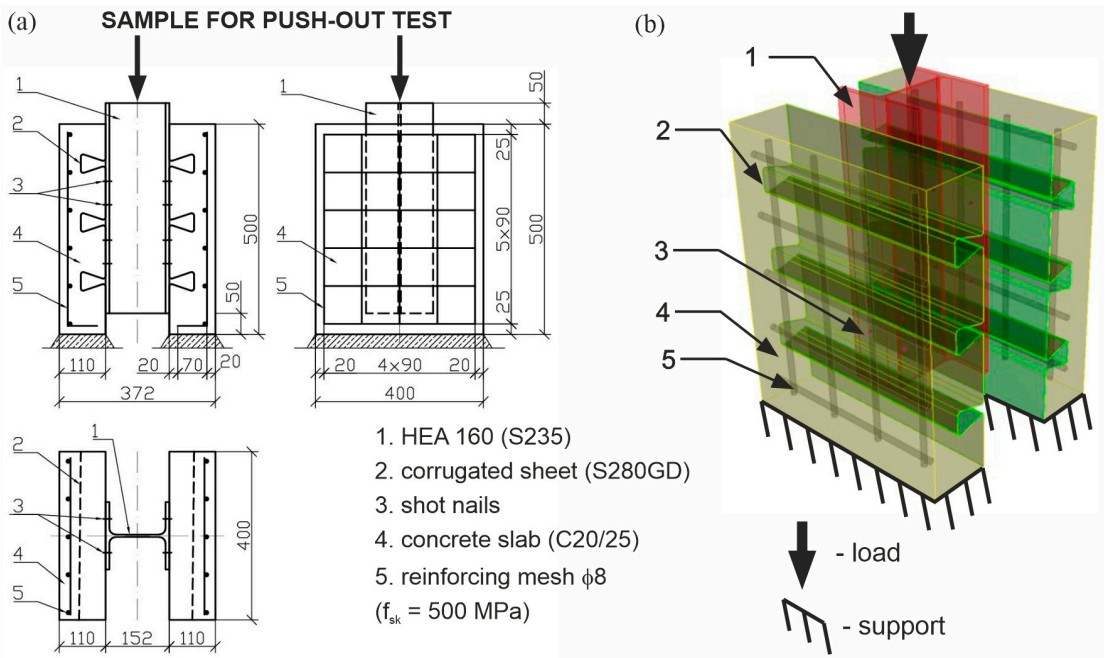

**Figure 3.** Sample for push-out test: (**a**) technical drawing; and (**b**) 3D view.

The geometry of the specimens for the shear tests, taking into account the dimensions of the concrete slab and the reinforcement mesh, is shown in Figure 3. The reinforced concrete slabs were made of C20/25 concrete class. The thickness of the slab (above the folds of the sheet) was 51 mm, so that the total thickness of the slab was 110 mm. The width of the plates was 400 mm and their length was 500 mm. The reinforcement mesh was prepared from ϕ 8 mm bars made of steel with a characteristic yield strength $f_{sk}$ = 500 MPa and C ductility class according to EN 1992-1-1 [30]. The reinforcement bars were spaced every 90 mm. The used concrete cover shown in Figure 3 met the requirements with their minimum thickness of 15 mm.

It should be emphasized that the connector consisted of the corrugated metal sheet and several nails shot into the single fold of the sheet (Table 3). For each type of connector model, three nominally identical samples marked A, B, and C were made. To determine

the initial load to failure of the sample, a push-out test was performed on one preliminary sample, designated as model no. 1. The model was characterized by a fastener made of a 1.00 mm thick sheet and 4 pieces of nails arranged in a single corrugation of the sheet. The spacing of the driven nails, showing the view of the sample before concreting, is shown in Figure 4. The proper push-out tests of the fasteners were carried out on models no 2, 3, and 4. The model no. 2 (Figure 4a) was identical to model no. 1, i.e., it was made of connectors made of 1.00 mm thick and 4 nails in a single sheet fold. The slabs in the model no. 3 (Figure 4b) were poured over 1.00 mm thick metal sheet fixed to the I-beam with 2 pieces of nails in the single sheet fold. In model no. 4 (Figure 4b), a 1.25 mm thick sheet and 2 pieces of nails were used in the fold of the sheet.

**Table 3.** List of analyzed fasteners.

| Model of Connector | 1 | 2 | 3 | 4 |
|---|---|---|---|---|
| Sheet thickness, mm | 1.00 | 1.00 | 1.00 | 1.25 |
| Number of nails per fold $n_g$ | 4 | 4 | 2 | 2 |

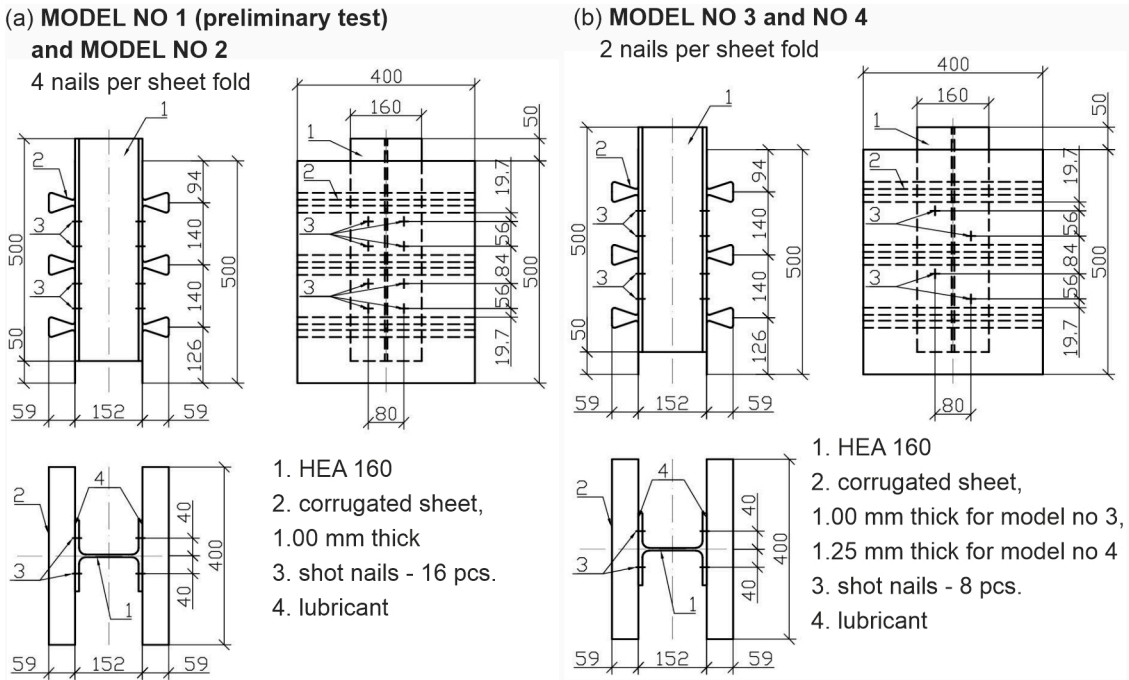

**Figure 4.** Sample for push-out test before concreting: (**a**) model no. 1 for preliminary tests and model no. 2; and (**b**) model no. 3 and 4, mm.

Each slab was concreted in a horizontal position, as performed in the practice of concreting composite beams (Figure 5). The samples were matured in an air environment. The push-out tests were carried out 28 days after the samples were made (Figure 6).

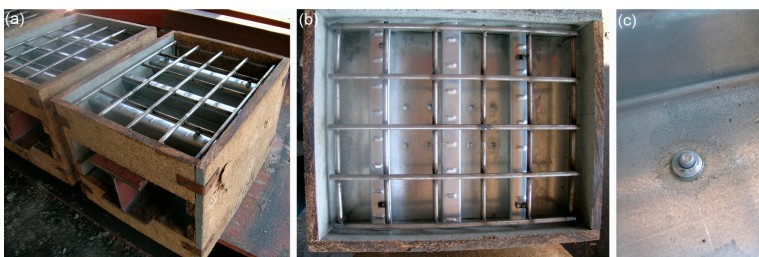

**Figure 5.** Samples prepared for concreting slabs in a horizontal position: (**a**) view; (**b**) view of the corrugation sheet and reinforcement; and (**c**) view of nail driven into the sheet.

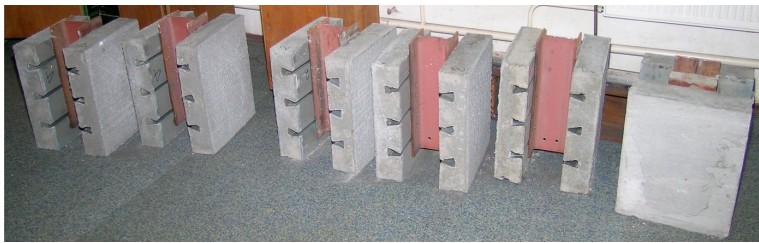

**Figure 6.** Samples for push-out tests.

The test procedure described in EN 1994-1-1 [20] was used. Therefore, the tests of each model of connector were carried out on three nominally identical samples. The test stand provided the ability to measure the load and displacement (longitudinal slip between the reinforced concrete slab and the steel cross-section) using four dial gauges. Sensors were positioned to measure the displacement at each connector as shown in Figure 7. Each sample was first loaded with 40% of the expected breaking load (maximum carried load) and then the sample was subjected to 25 load cycles between 5% and 40% of the expected breaking load. Subsequent load increments were introduced in such a way that failure did not occur in less than 15 min.

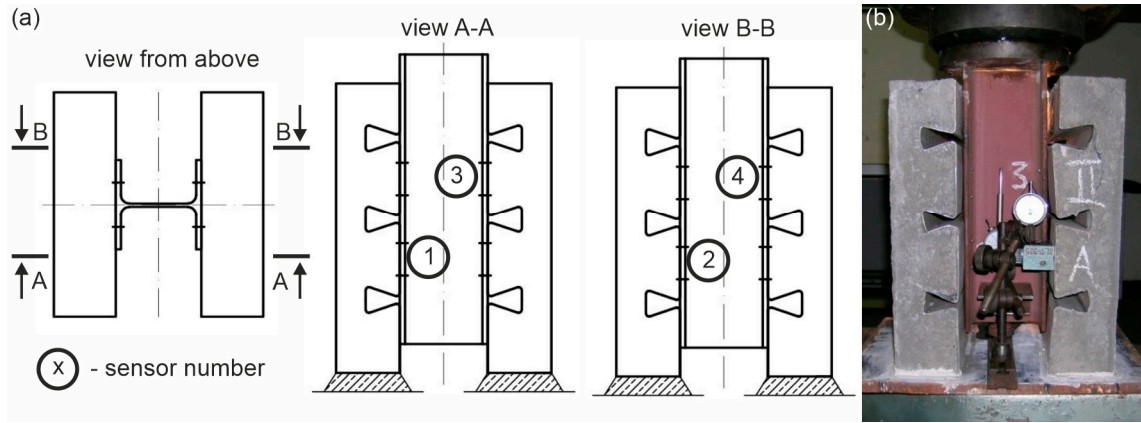

**Figure 7.** Arrangement of the dial gauges on the sample for the push-out test: (**a**) scheme; and (**b**) sample photo.

Therefore, the push-out test was performed on one preliminary specimen (model no. 1) to determine the initial failure load. As a result of the calculations carried out based on the material specification, it was estimated that the expected breaking force of model no. 1 was 300 kN. During the experiment, the sample was first loaded with the force in the range from 0 to 135 kN; then, it was subjected to 25 cycles of loading between 15 kN and 135 kN, and in the final phase of testing, it was loaded until destruction. The obtained breaking load in the preliminary test allowed the reduction in the load value for the samples in models no. 2–4. The cyclic loads for samples of models no. 2–4 were in the range of 10 kN–60 kN.

## 3. Results

As a result of the experimental tests, the mechanical properties of the materials and the shear load capacity of the connectors were determined.

### 3.1. Material Properties

The following results were obtained from the static tensile test of the steel:

- The yield strength of each samples marked $R_e$ in accordance with EN ISO 6892-1 [28];
- Nominal yield strength marked $f_y$ (according to EN 1994-1-1 [20]), which was taken as the mean value from experimental tests;

- Tensile strength of each sample marked $R_m$ according to [28];
- Nominal ultimate tensile strength marked $f_u$ according to [20], which was taken as the average value from experimental tests.

The results for the HEA 160 I-beam material used for the shear test specimens of the fasteners are shown in Table 4. The HEA 160 material meets the requirements of S235JR steel grade, as the average yield strength and tensile strength of 275 MPa and 405 MPa, respectively, meet the requirements of EN 10025-2 [27], which define the minimum yield strength for steel grade S235 at the level of 235 MPa and tensile strength in the range of 350–510 MPa.

**Table 4.** Results from static tensile tests of HEA 160 steel.

| Sample No. | Yield Strength $R_e$, MPa | Tensile Strength $R_m$, MPa | Relative Elongation A5, % |
|---|---|---|---|
| HEA 160.1 | 272 | 409 | 34.8 |
| HEA 160.2 | 249 | 386 | 35.6 |
| HEA 160.3 | 304 | 422 | 38.8 |
| Average $\bar{x}$: | $f_y = 275$ | $f_u = 405$ | 36.4 |

The material test results for the 1.00 and 1.25 mm thick corrugated sheets used to make the test specimens of the fasteners as well as for the composite beam are shown in Table 5. The average yield strength of the 1.0 mm and 1.25 mm metal sheet are, respectively, 300 and 304 MPa and they are higher than the required 280 MPa according to EN 10346 [26]. On the other hand, the average tensile strength of 1.0 mm and 1.25 mm thick sheets is 379 and 382 MPa, respectively, and is higher than the 360 MPa required by the standard [26]. Therefore, the material of metal sheet meets the requirements for steel grade S280GD.

**Table 5.** Results from the test of corrugated sheets.

| Nominal Thickness, mm | Sample No. | Yield Strength $R_e$, MPa | Tensile Strength $R_m$, MPa |
|---|---|---|---|
| 1.00 | T-59.1 | 303 | 384 |
| | T-59.2 | 301 | 378 |
| | T-59.3 | 297 | 374 |
| | Average $\bar{x}$: | $f_y = 300$ | $f_u = 379$ |
| 1.25 | T-60.1 | 301 | 382 |
| | T-60.2 | 305 | 381 |
| | T-60.3 | 307 | 383 |
| | Average $\bar{x}$: | $f_y = 304$ | $f_u = 382$ |

Test results of the strength class of the concrete used for the construction of connectors are presented in Table 6. The average characteristic compressive strength $f_{ck.cube}$ for the tested concretes is 33.1 and 32.2 MPa. In accordance with EN 206+A2 [29], all tested samples met the requirements for the concrete class C20/25.

**Table 6.** Test results of concrete class used for preliminary and main samples.

| Purpose of Concrete | Sample No. | Characteristic Compressive Strength $f_{ck.cube}$, MPa |
|---|---|---|
| Used for preliminary test | K11 | 33.0 |
| | K12 | 32.0 |
| | K13 | 34.2 |
| | Average $\bar{x}$: | 33.1 |
| Used for main tests | K21 | 31.9 |
| | K22 | 32.6 |
| | K23 | 32.1 |
| | Average $\bar{x}$: | 32.2 |

*3.2. Push-Out Test*

The failure modes, determination of load-bearing capacity, as well as slip assessment were presented.

### 3.2.1. Failure Modes

All samples with the connector made of 1.00 mm thick corrugated sheet and four driven nails (fastener model no. 2) failed, as shown in Figures 8–10. Initially, the connection between the steel beam and the concrete slab was fully rigid, which led to a significant increase in the load capacity of the connections. No damage of the samples was observed in the first or second phase of loading. When the load reached its shear strength limit, a vertical crack occurred in the concrete slab near the top fold of the sheet (Figure 8). Then, as the slip increased, the concrete slabs moved in such a way that the folds of the sheet approached each other. The deformation of the concrete slabs caused the deformation of the corrugated sheet and finally its partial separation from the concrete slab (Figure 9). At the end of the test, the concrete slab was carefully removed to analyze the condition of the material around the nails. The nails were still firmly embedded in the steel beam (Figure 10a). Displacements of the sheet folds and concrete resulted in the plasticization of the metal sheet around the driven nails. No damage of the steel section was observed. The failures occurred near the concrete, because the shear strength of the steel is much larger than that of the concrete. The results indicate that the shearing of the concrete slab was the dominant factor in the failure of the samples in which four nails were used in each corrugation of the sheet.

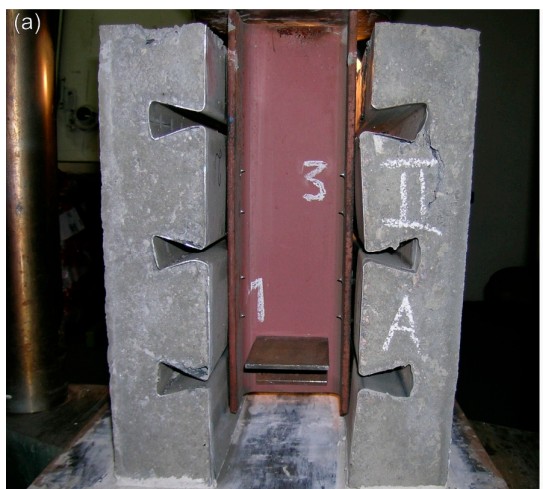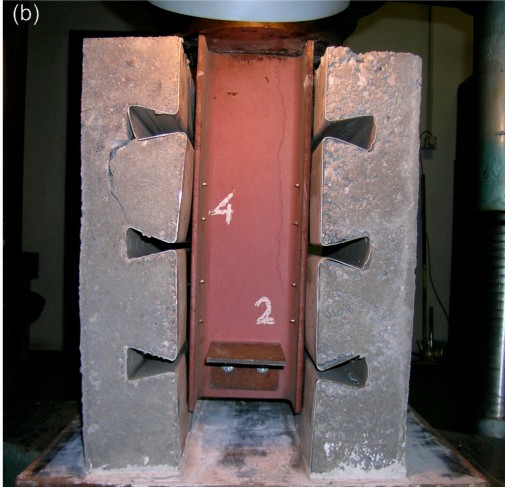

**Figure 8.** Failure of specimen from model no. 2 (consisted of 1.00 mm thick metal sheet and 4 shot nails) during shear test: (**a**) front view; and (**b**) rear view.

All samples of models no. 3 and 4, i.e., with fasteners in which two nails were driven into each fold of the sheet, failed in the same way, as shown in Figures 11 and 12. The load capacity of this type of connection increased rapidly at the beginning of the loading process. This was because the connection between the nail and the steel beam was quickly established, and the nail began to transfer the shear force to the corrugated sheet. No damage of the samples was observed in the first or second phase of loading. In the range of loads up to the values corresponding to the yield point of the fasteners, no cracks were observed in the tests either. Then, with the increase in slip, the tearing of the nails from the corrugated sheet and the concrete slab began to be observed (Figure 11). After the loading process was completed, no cracks were visible in the I-beam or on the surface of the concrete slab. The nails were still firmly embedded in the steel section. The nails protruding from the steel beam were deformed (Figure 12b). The corrugated sheet was deformed in the area of nails (Figure 12a). Most likely, the concrete was also crushed in the

area of the torn-out nail heads. The results indicate that the tearing out of the nails from the plasticized sheet was the dominant factor determining the destruction of the fasteners in which two nails were driven into the fold of the sheet.

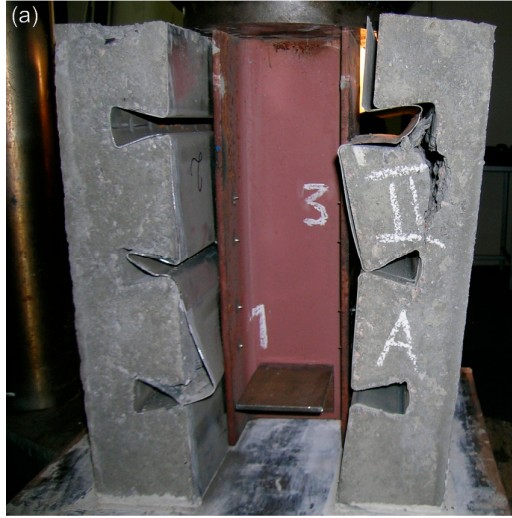
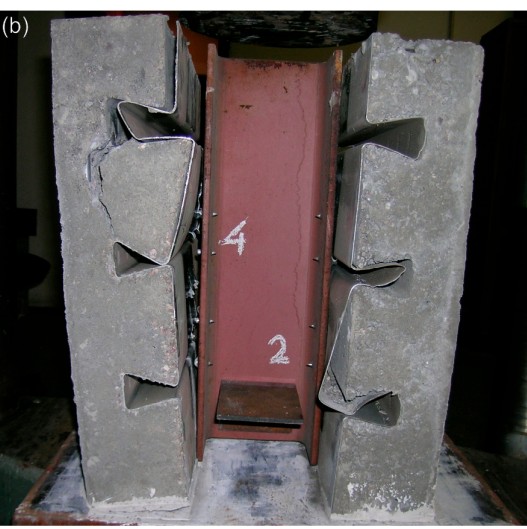

**Figure 9.** Failure of specimen from model no. 2 (consisted of 1.00 mm thick metal sheet and 4 shot nails) after shear test: (**a**) front view; and (**b**) rear view.

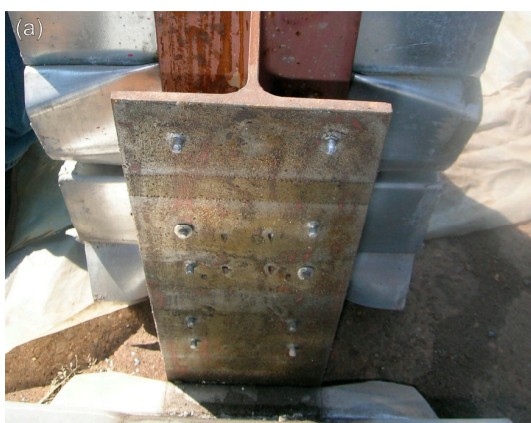
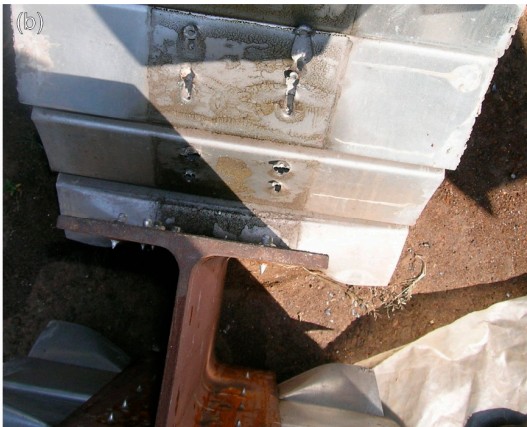

**Figure 10.** Failure of specimen from model no. 2 (consisted of 1.00 mm thick metal sheet and 4 shot nails) after separating concrete slab from steel beam: (**a**) steel beam view; and (**b**) corrugated sheet view.

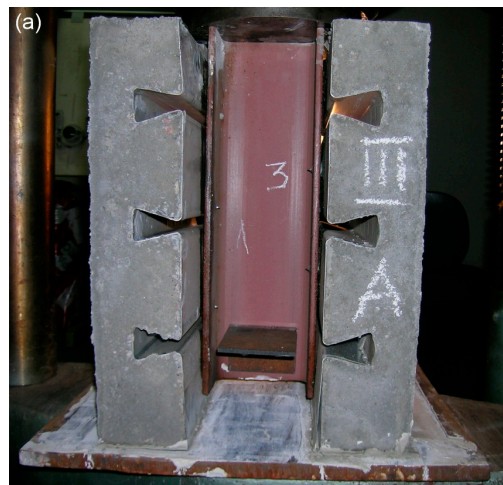
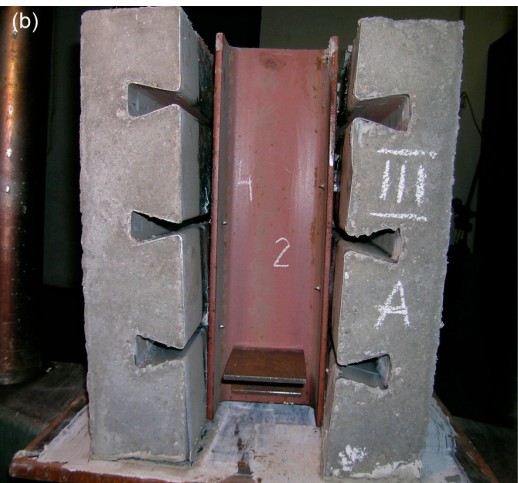

**Figure 11.** Failure mode of model no. 3 and 4 (consisted of 2 shot nails): (**a**) front view; and (**b**) rear view.

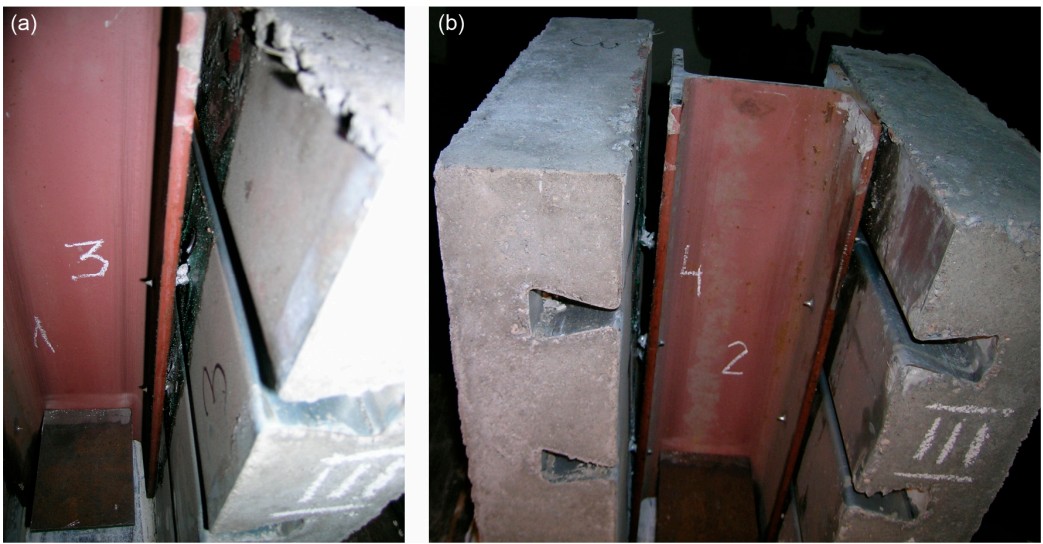

**Figure 12.** Failure mode of model no. 3 and 4 (consisted of 2 shot nails): (**a**) front view; and (**b**) rear view.

### 3.2.2. Mechanical Properties

Displacement–force diagrams for all analyzed samples are shown in Figure 13. Based on the results from the preliminary test (Figure 13a), the maximum force of 210 kN was determined, transmitted by the model with fasteners in the form of sheets with a thickness of 1.00 mm and four nails. Thanks to this, the characteristic values of 5% (10 kN) and 40% (80 kN) of the expected failure force of the samples in model no. 2, which were built of nominally identical connectors as in model 1, were determined.

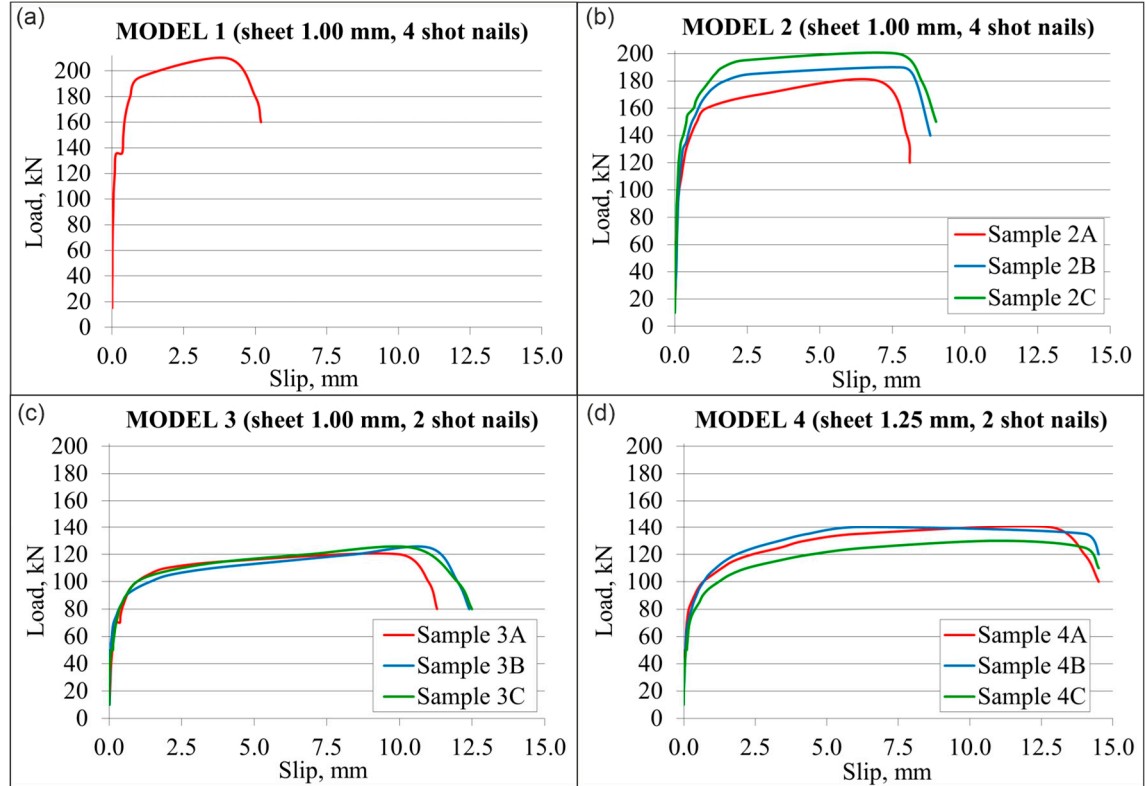

**Figure 13.** Load–slip curves from push-out tests: (**a**) model no. 1; (**b**) model no. 2; (**c**) model no. 3; and (**d**) model no. 4.

The maximum shear forces transferred by the samples of model no. 2 are 180, 190, and 200 kN. The displacement–force diagrams (Figure 13b) show that the curves for individual samples coincide in the elastic range, ranging from 0 to about 100–120 kN. After exceeding the yield point, the curves cease to coincide and their curvature can be observed. After exceeding the forces by about 160 kN, 180 kN, and 195 kN by samples 2A, 2B, 2C, respectively, there is a significant increase in displacement (by about 5.0 mm) in relation to a small increase in load (by about 20 kN). This phenomenon occurs until the forces destroying the sample are reached. The displacement of samples of model no. 2 at the moment of reaching the maximum load was in the range of 7.0–7.82 mm.

Analyzing the results from the push-out tests of model no. 3 (Figure 13c), in which fasteners made of sheet, 1.00 mm thick, and 2 nails were used, the convergence of the displacement–force curves was observed almost to the breaking strength of the samples. The maximum shear force transmitted by specimen 3A is 120 kN, while the maximum shear force transmitted by specimens the 3B and 3C is 125 kN. The shape of the displacement–force curves is similar to the curves in model no. 2. Initially, an increase in load causes a linear increase in displacement until the yield point is reached, corresponding to a load of about 60 kN. Further increasing the load causes the displacement to increase nonlinearly up to the breaking point. The displacement of model no. 3 samples at the moment of reaching the maximum load was in the range of 11.3–12.5 mm.

The results for model no. 4 samples built with fasteners in the form of 1.25 mm sheet and 2 pcs. of nails are shown in Figure 13d. The maximum shear forces transferred by samples 4A and 4B are 140 kN, and the breaking force of sample 4C is 130 kN. The displacement–force diagrams show that the curves coincide in the elastic range, i.e., up to the load value of about 70 kN. After exceeding the yield point, the curves cease to coincide, and their curvature can be observed. Then, after exceeding the force of 110–120 kN, there is a significant increase in displacement (by about 10.0 mm) in relation to a small increase in load (by about 20 kN). The displacement of model no. 4 samples at the moment of reaching the maximum load is in the range of 11.0–12.8 mm.

3.2.3. Determination of Load-Bearing Capacity

Connectors for composite structures should transfer the longitudinal delamination forces between the concrete and steel components and should prevent the separation of the concrete and steel components. To meet these requirements, the characteristic and design resistance of the fasteners must be determined. The characteristic and design resistance of the fasteners was determined on the basis of the guidelines of EN 1994-1-1 [20]. Therefore, in the first place, the breaking load of each fastener $P_{1Rk}$ was determined, calculated as the quotient of the total breaking force of the sample $P_{0Rk}$ and the number of fasteners $n_r$:

$$P_{1Rk} = \frac{P_{0Rk}}{n_r} \tag{1}$$

where fastener was meant to be one fold of corrugated metal sheet and nails in such a fold.

Then, for each of the models, the arithmetic mean of the breaking load of the connector $P_{S.Rk}$ and the deviation from the average value for each of the samples were determined. Since three nominally identical samples were tested according to the standard [20] and the deviation of the individual test result from the average value obtained from all tests does not exceed 10% (Tables 7–9), the load capacity of the fasteners was determined as follows:

- Characteristic resistance $P_{Rk}$ as the minimum breaking load reduced by 10%:

$$P_{Rk} = P_{1Rk} \times 90\% \tag{2}$$

- Design resistance $P_{Rd}$ calculated from the formula:

$$P_{Rd} = \frac{f_u}{f_{ut}} \frac{P_{Rk}}{\gamma_v} \tag{3}$$

where the following definitions apply:

$f_u$—the minimum specified ultimate strength of the connector material;
$f_{ut}$—the actual ultimate strength of the connector material;
$\gamma_v$—the partial safety factor for shear connection equal to 1.25.

The procedure for evaluating the results presented in EN 1994-1-1 [20] mainly applies to the testing of typical headed studs connectors made of nominally homogeneous material. On the other hand, the analyzed fasteners are made of two materials: the sheet material and the material of the driven nail. Currently, there is no minimum specified ultimate strength $f_u$ of the two-component connector. Therefore, the ratio of the minimum specified ultimate strength of the connector material $f_u$ and the actual ultimate strength of the connector material $f_{ut}$ equal to 1.0 was adopted for the calculation of the design resistance of the connectors $P_{Rd}$.

In model no. 2 (Table 7), the total destructive forces $P_{0Rk}$ of individual samples are 180 kN, 190 kN, and 200 kN. Hence, the destructive forces of the connectors $P_{1Rk}$ are equal to 45.00 kN, 47.50 kN, and 50.00 kN. Hence, the average value of the breaking load of the connector $P_{S.Rk}$ is 47.50 kN, and the deviations from the average value of 0% and 5.3% are less than 10%. Therefore, the characteristic resistance $P_{Rd}$ and the design resistance $P_{Rd}$ of the fastener made of 1.00 mm thick sheet and 4 nails are 40.50 kN and 38.00 kN, respectively.

**Table 7.** Data to determine the capacity of the connector model no. 2.

| Sample | 2A | 2B | 2C |
|---|---|---|---|
| Destructive force $P_{0Rk}$, kN | 180 | 190 | 200 |
| Number of connectors in the sample $n_r$ | | 4 | |
| Sheet thickness, mm | | 1.00 | |
| Number of nails per one-fold (in one connector) $n_g$ | | 4 | |
| Destructive load of the fastener $P_{1Rk}$, kN | 45.00 | 47.50 | 50.00 |
| Average value of the breaking load of the fastener $P_{S.Rk}$, kN | | 47.50 | |
| Deviation from the mean value, % | 5.3% | 0.0% | 5.3% |
| Characteristic load capacity of the fastener $P_{Rk}$, kN | | 40.50 | |
| Design resistance of the connector $P_{Rd}$, kN | | 38.00 | |

In model no 3 (Table 8), with fasteners made of 1.00 mm thick sheet and 2 nails, the total failure forces $P_{0Rk}$ of the individual samples are 120 kN, 125 kN, and 125 kN. The destructive forces of connectors $P_{1Rk}$ are equal to 30.00 kN and 31.25 kN. The average value of the breaking load of the connector $P_{S.Rk}$ is 30.83 kN, and the deviations from the average value of 2.70% and 1.35% are less than 10%. The characteristic resistance $P_{Rk}$ and the design resistance $P_{Rd}$ of the connector in the model no. 3 are 27.00 kN and 24.67 kN, respectively.

**Table 8.** Data to determine the capacity of the connector model no. 3.

| Sample | 3A | 3B | 3C |
|---|---|---|---|
| Destructive force $P_{0Rk}$, kN | 120 | 125 | 125 |
| Number of connectors in the sample $n_r$ | | 4 | |
| Sheet thickness, mm | | 1.00 | |
| Number of nails per one fold (in one connector) $n_g$ | | 2 | |
| Destructive load of the fastener $P_{1Rk}$, kN | 30.00 | 31.25 | 31.25 |
| Average value of the breaking load of the fastener $P_{S.Rk}$, kN | | 30.83 | |
| Deviation from the mean value, % | 2.70% | 1.35% | 1.35% |
| Characteristic load capacity of the fastener $P_{Rk}$, kN | | 27.00 | |
| Design resistance of the connector $P_{Rd}$, kN | | 24.67 | |

In model no 4 (Table 9) made with fasteners of 1.25 mm thick sheet and 2 nails, the total breaking forces $P_{0Rk}$ of the individual samples are 140 kN, 140 kN, and 130 kN. The destructive forces of connectors $P_{1Rk}$ are equal to 35.00 kN and 32.50 kN. Hence, the average value of the breaking load of the connector $P_{S.Rk}$ is 34.17 kN, and the deviations from the average values of 2.44% and 4.88% are less than 10%. The characteristic resistance $P_{Rk}$ and the design resistance $P_{Rd}$ of the connector in model no 4 are 29.25 kN and 27.33 kN, respectively.

**Table 9.** Data to determine the capacity of the connector model no. 4.

| Sample | 4A | 4B | 4C |
|---|---|---|---|
| Destructive force $P_{0Rk}$, kN | 140 | 140 | 130 |
| Number of connectors in the sample $n_r$ | | 4 | |
| Sheet thickness, mm | | 1.25 | |
| Number of nails per one fold (in one connector) $n_g$ | | 2 | |
| Destructive load of the fastener $P_{1Rk}$, kN | 35.00 | 35.00 | 32.50 |
| Average value of the breaking load of the fastener $P_{S.Rk}$, kN | | 34.17 | |
| Deviation from the mean value, % | 2.44% | 2.44% | 4.88% |
| Characteristic load capacity of the fastener $P_{Rk}$, kN | | 29.25 | |
| Design resistance of the connector $P_{Rd}$, kN | | 27.33 | |

### 3.2.4. Slip

Shear connectors must have sufficient deformability to justify an inelastic shear redistribution. The fastener may be considered ductile if its characteristic sliding capacity $\delta_{uk}$ is at least 6.0 mm [20]. The slip capacity $\delta_u$ shall be taken equal to the slip measured at the maximum test load. According to the requirements assumed in EN 1994-1-1 standard [20], the characteristic slip capacity $\delta_{uk}$ was assumed to be equal to the minimum value $\delta_u$ among the tested samples, reduced by 10%. Data to determine the characteristic slip capacity of analysed samples is presented in Table 10.

**Table 10.** Data to determine the characteristic slip capacity.

| Model of Connector | 2 | 3 | 4 |
|---|---|---|---|
| Minimum destructive load of the fastener $P_{1Rk.min}$, kN | 45.00 | 30.00 | 32.50 |
| $P_{1Rk} \cdot 90\%$, kN | 40.50 | 27.00 | 29.25 |
| Characteristic ability to slip $\delta_{uk}$, mm | 7.70 | 10.77 | 14.40 |

Among the samples from model no. 2, the smallest destructive force of the connector was transferred by sample A. Therefore, Figure 14a shows a slip-force graph for a single connector from sample 2A. The graph shows the characteristic slip capacity $\delta_{uk}$ equal to 7.70 mm, corresponding to 90% of the fastener failure force. Detailed values are presented in Figure 14a.

Figure 14b shows the slip–force diagram for a single fastener from sample 3A, because among the samples of model no. 3, sample A showed the lowest destructive force of the fastener. Analyzing the results of the samples from model no. 4, the smallest destructive force of the fastener was transferred by sample C.

Based on Figure 14c, the characteristic sliding ability $\delta_{uk}$ equal to 14.40 mm was determined. Since the characteristic slip capacity for all models is greater than 6.0 mm, the analyzed fasteners were considered ductile.

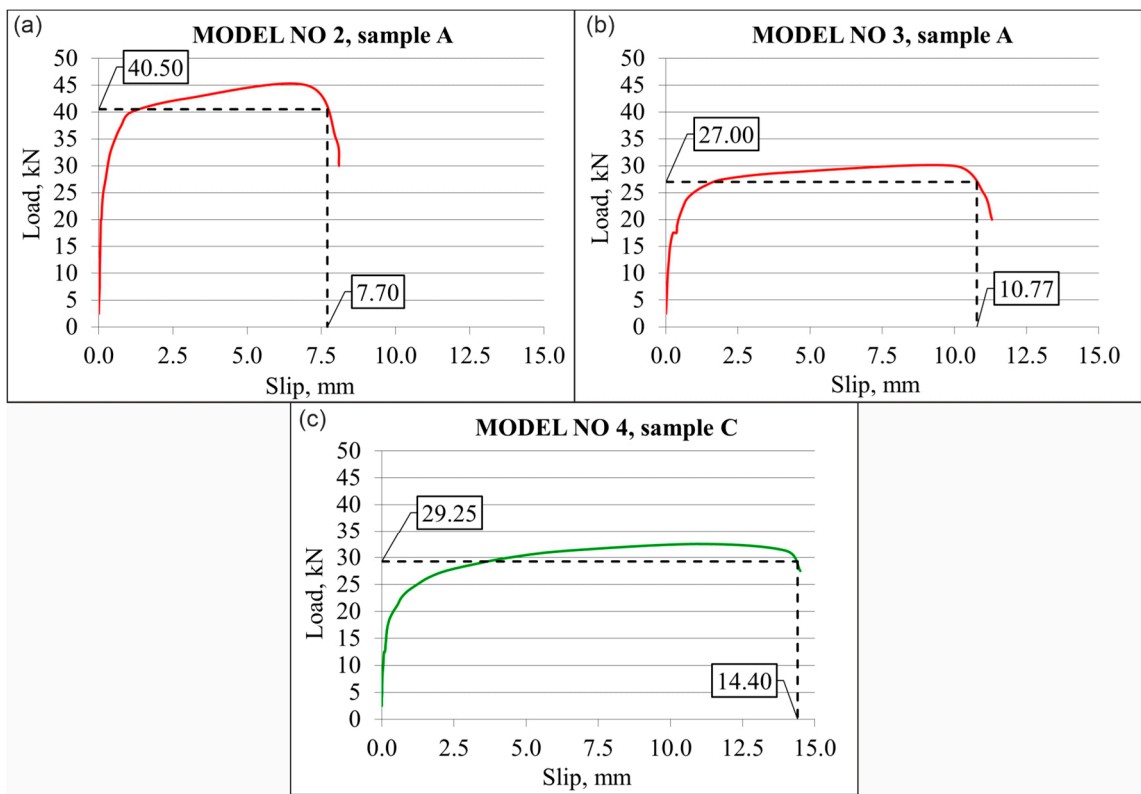

**Figure 14.** Slip–force diagram to determine the slip of fastener for (**a**) sample 2A (1.00 mm thick sheet and 4 nails); (**b**) sample 3A (1.00 mm thick sheet and 2 nails); and (**c**) 4C samples (1.25 mm thick sheet and 2 nails.

### 4. Discussion

The summary of the results from the experimental tests of the fasteners is presented in Table 11. The average value of the breaking load $P_{S.Rk}$ of the fastener in model no. 2, which was built using fasteners in the form of the corrugated sheet with a thickness of 1.00 mm and 4 nails, obtained experimentally is 47.50 kN. Reducing the number of nails by 50%, i.e., from 4 to 2, resulted in reducing the average breaking load $P_{S.Rk}$ of the fastener by 35%, because the average breaking load of the fastener in model no. 3 in which fasteners made of 1.0 mm thick corrugated sheet and 2 nails were used, and experimentally obtained to be 30.83 kN. However, increasing the sheet thickness from 1.00 to 1.25 mm, i.e., by 25% (while maintaining 2 nails), resulted in a return of the average breaking load $P_{S.Rk}$ of the connector by only 11%, because the average breaking load $P_{S.Rk}$ of the connector in model 4, in which fasteners were made of corrugated sheet metal with a thickness of 1.25 mm and 2 nails, obtaining the value of 34.17 kN.

**Table 11.** Summary of the results from the experimental tests of fasteners.

| Model of Connector | 2 | 3 | 4 |
|---|---|---|---|
| Average value of the breaking load of the fastener $P_{S.Rk}$, kN | 47.50 | 30.83 | 34.17 |
| Characteristic load capacity of the fastener $P_{Rk}$, kN | 40.50 | 27.00 | 29.25 |
| Design resistance of the connector $P_{Rd}$, kN | 38.00 | 24.67 | 27.33 |
| Slip capacity $\delta_{uk}$, mm | 7.70 | 10.77 | 14.40 |

Comparing the weight of the adopted fastener, which is affected by the thickness of the sheet and the number of nails, as well as taking into account the labor consumption of driving two and four nails, the fastener from model no 3, which was made of 1.00 mm thick corrugated sheet and two nails, was considered the most effective. It cannot be said that

this is the best solution, because the selection of fasteners in composite structures always depends on the acting load. Therefore, in places where a greater load capacity is required, it was decided that a better solution was to increase the number of nails to four than to increase the thickness of the sheet. This is supported by the fact that the fastener with four nails carries a 35% higher load than the fastener with 2 nails but increasing the sheet thickness from 1.00 to 1.25 mm results in the increase in load by only 11%. It should be remembered that the labor consumption of using a thicker sheet metal is practically nil compared to shooting 50% more fasteners. However, in one structural member, it is possible to use fasteners with two and four nails in such a way as to optimize the fasteners and increase the number of nails only locally, in areas where greater load capacity is required.

Among all analyzed connector models, the maximum deviation from the average value is equal to 5.3%. This value was considered mean, because EN 1994-1-1 [20] allows deviations of up to 10%. The reason for the deviation of the results of individual samples from the average value are inaccuracies in the execution of materials, components of the samples (I-beam, sheet metal, nails) and inaccuracies in the execution of the samples subjected to the shear test.

Ductile connectors are characterized by sufficient deformability to justify the perfectly plastic behavior of the connection in the considered structure. As shown in Table 11, all analyzed fasteners were considered ductile as they had the characteristic slip capacity $\delta_{uk}$ of at least 6.0 mm. The connector in model no. 2 showed the lowest slip capacity, because it is 7.70 mm. Reducing the number of nails from four to two resulted in a 29% increase in slip, because in model no. 3, it was 10.77 mm. The adoption of two nails and increasing the thickness of the sheet from 1.00 to 1.25 mm resulted in an increase in the slip by 25%, because in model no. 4, it was 14.40 mm. Therefore, the slip of model no. 2 is 47% higher than that of model no. 4. Reducing the number of nails (while maintaining the nominally identical sheet thickness) increases the slip, because the stiffness of the fastener decreases. An increase in the thickness of the sheet (while maintaining the same number of nails) also increases the slip capacity. This is because the use of a thicker sheet causes the nails to become more loaded than in the case of a thinner sheet.

Push-out test is recommended by the EN 1994-1-1 [20] standard to verify the behavior of new types of connectors (other than headed studs) that can be used in real-world conditions. The recommended test is intended to replicate the pure shear performance of the connectors. However, it should be taken into account that, under real conditions, connectors used in steel–concrete composite beams may be sheared as a result of the movement of the slab relative to the beam, as well as compressed and sometimes even stretched. Additionally, the push-out shear test may contain potential biases, because the movement of the I-beam, in addition to shearing the connectors, may also cause the slabs to detach due to the resulting lateral forces.

The proposed solution of fastener for composite structures met the assumed load capacity expectations and it is easy in fabrication. The use of traditional headed stud fasteners required special mounting equipment. Such devices require high-current electrical power, which negatively affects the environment. At the same time, resourceful people with appropriate qualifications are necessary for the installation of headed studs. These limitations are eliminated in the proposed connector solution in the form of the corrugated sheet fastened with shot nails. A device for driving nails is available in every construction factory and even on construction sites. This is due to the universal use of driven nails, which are used in various types of building structures. These types of devices do not require electricity, which reduces the negative impact on the environment, unlike traditional solutions requiring welding. To make a steel–concrete composite beam with traditional headed connectors, the corrugated sheet is required. This sheet is attached to the I-beam with nails, and headed studs are needed to connect the sheet with the concrete slab. In the proposed solution, head studs were eliminated. This contributed to reducing the weight of the composite beam and shortening the time needed to manufacture the beam, because one operation was eliminated, i.e., the assembly of studs. Reducing the assembly time also

translates into lower production costs. In both solutions, the weight of the sheet metal and nails as well as the time necessary to attach the sheet to the I-beam are estimated to be similar.

The conducted research proved the thesis that it is possible to provide the steel–concrete connection without the use of additional connecting components. In the tested connector, the friction joint is created thanks to the use of especially shaped folds of the sheet in the so-called dovetail. The shape of such a cross-section improves the stability of the sheet, as well as acts as hooks connecting to the concrete, preventing the global buckling of the slab. An additional advantage of this sheet is the ability to easily suspend finishing elements by using dedicated hangers fixed in the fold. The formwork is not performed during the manufacturing of steel–concrete composite structures with the analyzed connector, because the corrugated sheet performs this function.

The proposed solution is not an alternative to headed studs, but it a solution that can be used in other applications. Design the resistance of the headed stud is about 60 kN. Design resistance of the connector made of dovetail sheet and shot nails is about 30 kN. Therefore, the proposed connectors are characterized by a relatively low load-bearing capacity. Hence, they are a very good solution for small utility public buildings. It is true that the dimensions of the building itself do not matter. The spans of the beams and their load are important. These features translate into the values of internal forces in the beams, which affect the thickness of the slab. Engineering experience shows that small utility public buildings are characterized by relatively small beam spans and thin slabs. In such cases, the use of classic headed connectors results in their underutilization, because the connectors have a much higher load capacity than required.

The restrictions on the use of connectors made of dovetail sheet and shot nails are the same as the restrictions for whole types of ceilings made of corrugated sheets. This means that such ceilings should not be installed outdoors due to the presence of moisture contributing to corrosion of the sheet metal. Given the long-term durability of these connections, corrosion may also occur at the interface between the nails and sheet metal. The nails are galvanized, so even though some part of the nails protrude below the beam, they should not corrode. Compared to headed studs, the possibility of concrete chipping is significantly reduced. The chipping of concrete around the studs is completely eliminated because there are no pins in the proposed solution. However, the possibility of concrete chipping in the area of the folds of the sheets is similar in all ceilings made on corrugated sheets. The steel–composite composite beams with connectors made of dovetail sheet and shot-in nails should be used in environmentally controlled buildings, for example, in utility public buildings or in light industry, e.g., in the automotive, where very dry conditions prevail.

In the future, it is planned that experimental tests will be carried out on a composite steel–concrete beam made using the connector made of corrugated sheet metal and nails. Then, the numerical models of such structures will be developed. The sheet, nails and concrete material, the number and arrangement of nails, and the thickness of the sheet metal will be parameterized.

## 5. Conclusions

1. Based on the experimental tests, it was found that the proposed solution of fasteners made of corrugated sheet and shot nails can be used as the connector for steel–concrete composite floor beams.
2. Each type of analyzed fasteners was found to be ductile based on the criterion presented in Eurocode 4 standard.
3. The sheet thickness and the number of nails determine the load capacity of the fasteners. The load-bearing capacity of the fastener can be adjusted by changing the corrugated sheet thickness and changing the number of nails shot in the single fold of the sheet.
4. The breaking load of the connector made of 1.00 mm thick corrugated sheet and 4 pcs. of shot nails is 47.50 kN. The connector made of 1.00 mm thick sheet and two pieces of

nails is characterized by breaking load of 30.83 kN. The breaking load of the connector made of 1.25 mm thick sheet and two pieces of nails is 34.17 kN.

5.  It is planned that experimental tests of the steel–concrete composite beam will be carried out with the use of tested connectors made of corrugated sheet in the shape of a dovetail and shot nails. At the same time, it is planned to make a parameterized numerical model that would allow one to optimize the steel–concrete composite beam, especially the geometry of the sheet as well as the number and spacing of the nails.

**Author Contributions:** Conceptualization, P.K.; methodology, A.D. and P.L.; formal analysis, A.D.; investigation, A.D. and P.K.; resources, P.K.; data curation, P.K.; writing—original draft preparation, A.D., P.L. and P.K.; writing—review and editing, S.G.; visualization, A.D.; supervision, P.K.; project administration, A.D.; funding acquisition, P.K. All authors have read and agreed to the published version of the manuscript.

**Funding:** This research received no external funding.

**Institutional Review Board Statement:** Not applicable.

**Informed Consent Statement:** Not applicable.

**Data Availability Statement:** The raw data supporting the conclusions of this article will be made available by the authors on request.

**Conflicts of Interest:** The authors declare no conflicts of interest.

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
