# Peer review of "Study of Innovative Connector for Steel–Concrete Composite Structures"

_applsci, doi:10.3390/app14073003_

Round 1

Reviewer 1 Report

Comments and Suggestions for Authors

The paper provides a comprehensive introduction to the subject of steel-concrete composite beams, outlining the current structural solutions and the need for improved connectors. It reviews various existing methods, highlighting their limitations and setting a solid foundation for the proposed research. The references cited are relevant and support the discussion effectively.

The research design is well-structured, focusing on the innovative connector's experimental analysis. The methods are described in detail, including the materials used, the design of the connectors, and the push-out test procedures, allowing for reproducibility.

Results are presented clearly, with data on the mechanical properties of materials, failure modes, load-bearing capacities, and slip capacities. The conclusions drawn are well-supported by the experimental outcomes, indicating the potential of the proposed connector in enhancing the load-bearing capacity and ductility of steel-concrete composite beams.

The quality of English language is good, with technical terms appropriately used, making the paper accessible to professionals in the field. Overall, the paper appears to contribute valuable insights to the domain of civil engineering, particularly in the context of steel-concrete composite structures.

Author Response

We are very much thankful to the reviewers for their deep and thorough review of the manuscript entitled “Study of innovative connector for steel-concrete composite structures”.

Reviewer 2 Report

Comments and Suggestions for Authors

The manuscript presents an innovative approach to steel-concrete composite beam connectors utilizing corrugated sheets and shot nails. This method proposes a feasible alternative to conventional connectors, offering ease of fabrication and versatility in application, particularly for small utility public buildings. Experimental tests validate the ductility and load-bearing capacity of the fasteners, aligning with Eurocode 4 standards. The study is well-structured, providing a comprehensive analysis of material properties, failure modes, and load capacities. However, the paper would benefit significantly from addressing certain technical and methodological aspects to enhance its contribution to the field.

  1. Could the authors elaborate on the selection criteria for the materials used in the experiments? Specifically, how does the choice of S280GD steel and the specified nail dimensions contribute to the overall performance of the connector?
  2. The push-out tests provide essential data, but could the authors discuss any limitations or potential biases in these tests? How representative are these tests of real-world conditions in steel-concrete composite construction?
  3. The manuscript briefly mentions traditional headed studs but lacks a detailed comparative analysis. Could the authors provide a more detailed comparison, including cost, installation time, and performance differences between the proposed connectors and conventional solutions?
  4. The paper mentions planned numerical modeling for optimization purposes. Could the authors share preliminary thoughts or frameworks on how they intend to model these connectors? What specific aspects of the connector or beam behavior are they aiming to optimize?
  5. While the mechanical properties and immediate load-bearing capacities are well-addressed, there is little discussion on the long-term durability of these connectors. Could the authors speculate or provide data on how these connectors might perform over time, especially under varying environmental conditions?
  6. The ease of fabrication and the use of readily available materials are highlighted as advantages. Could the authors elaborate on the economic benefits, including potential cost savings, and any environmental impacts or benefits associated with using these connectors?
  7. The focus is on small utility public buildings. Could the authors discuss the potential applicability of their findings to other types of structures or larger-scale projects? Are there any inherent limitations in the scalability of this connector system?

Author Response

We are very much thankful to the reviewers for their deep and thorough review of the manuscript entitled “Study of innovative connector for steel-concrete composite structures”. We have revised our present research paper in the light of your useful suggestions and comments. I hope our revision has improved the paper to a level of your satisfaction. Our answers to your specific comments/suggestions/queries are as follows.

Reviewer 3 Report

Comments and Suggestions for Authors

Dear authors, 

I suggest including a comparison with existing models and systems. It is an experimental work carried out correctly but the progress achieved is not highlighted. Is it better than what exists?

Because it is cheaper, more effective, easier to execute,...

With current models, is it valid?

In this paper, this are the most important questions:   1. What parts do you consider original or relevant for the field? What specific gap in the field does the paper address?
2. What does it add to the subject area compared with other published material?

The study is well done but in my opinion, it is not clear the progress achieved.

Author Response

(The authors gave the same response as above.)

Round 2

Reviewer 2 Report

Comments and Suggestions for Authors

The revised version can be accepetd for publication.

Reviewer 3 Report

Comments and Suggestions for Authors

Dear authors, 

I appreciate the effort to correct and improve the article based on the comments received.

Everything is explained.

Comments on the Quality of English Language

English should be revised, there are some mistakes, especially in new contributions.